# Quantifying fish range shifts across poorly defined management boundaries

Juliano Palacios-Abrantes[1¤]*, Scott Crosson[2], Chris Dumas[3], Rod Fujita[4],
Arielle Levine[5], Catherine Longo[6], Olaf P. Jensen[1]

1 Center for Limnology, University of Wisconsin-Madison, Madison, WI, United States of America, 2 NOAA Southeast Fisheries Science Center, Miami, FL, United States of America, 3 Department of Environmental Sciences, University of North Carolina Wilmington, Wilmington, NC, United States of America, 4 Environmental Defense Fund, San Francisco, CA, United States of America, 5 Department of Geography, San Diego State University, San Diego, CA, United States of America, 6 Science & Standards, Marine Stewardship Council, London, United Kingdom

¤ Current address: Institute for the Oceans and Fisheries, The University of British Columbia, Vancouver, BC, Canada
* j.palacios@oceans.ubc.ca

**Data Availability Statement:** All primary data and code to generate results and figures can be found on GitHub (https://github.com/jepa/AcrossBoundaries). Secondary data used in this

## Abstract

Management regimes of marine resources that rely on spatial boundaries might be poorly adapted to climate change shifts in species distributions. This is of specific concern for the management of fish stocks that cross management jurisdictions, known as shared stocks. Transitioning to dynamic rules in spatial management has been suggested as a solution for mismatches between species distributions and the spatial boundaries. However, in many cases spatial boundaries are not clearly drawn, hampering such transitions. Here, we use black sea bass (*Centropristis striata*), summer flounder (*Paralichthys dentatus*) and scup (*Stenotomus chrysops*) as case studies to explore different approaches to designing spatial regulatory units to facilitate the adaptation of fisheries management to shifting distributions of shared stocks. First, we determine the yearly distribution of each stock within the United States Exclusive Economic Zone from 1951 to 2019 during Fall and Spring sampling seasons. Second, we explore two approaches for drawing regulatory units based on state waters and historical landings. Finally, we estimate each state's proportion of the stock's distribution and compare historical and recent values. We show that the distribution of all three stocks has changed relative to the years used to determine the current quota allocation across states, with an overall gain for central-northern states at the expense of the southern-most states. In terms of the distribution of allocation, we find that, while seasonal differences exist, the biggest differences in the proportion of the stock spatial distribution attributed to each state come from the method for designing regulatory units. Here, we show that the method used to define allocation units can have meaningful impacts on resulting adaptive policy. As climate change-driven conflicts in fishing resource allocation are expected to increase and deepen around the world, we provide a replicable approach to make an informed and transparent choice to support data-driven decision-making.

research are publicly available at Ocean Adapt
(https://oceanadapt.rutgers.edu/) and the Atlantic
Coastal Cooperative Statistics Program (ACCSP -
https://www.accsp.org).

**Funding:** This work was funded by the Lenfest
Ocean Program (https://www.lenfestocean.org/).
The funders had no role in study design, data
collection and analysis, decision to publish, or
preparation of the manuscript.

## Introduction

Range shifts are among the most documented responses of marine fauna to the effects of climate change in the ocean [1]. From zooplankton to bony fishes to seabirds, some distributions of marine fauna have expanded up to 800 km per decade, while others have contracted by about 200 km per decade [1]. Such change is often due to marine species following a poleward temperature gradient in the search for colder waters although in some cases, species are responding to local environmental factors, such as upwelling systems [2], or moving to deeper colder waters [3]. Regardless of the shift direction, changes in species distributions are projected to continue in the coming years [4], even if nations enact strong climate change mitigation measures (i.e., limiting increased global temperature to 1.5°C) as proposed by the Paris Agreement [5].

Ocean management tools at all levels need to incorporate adaptive methods to cope with shifting species distributions [6, 7]. For example, marine reserves intended to protect species within their boundaries will need to be composed of networks or be dynamic in nature [8] as part of a strategy to be climate resilient [9]. Moreover, fishing quota allocations based on historical distributions will most likely be outdated as climate changes shift species distributions requiring new allocation methods to follow the fish [10, 11]. As such, shifts in species distribution will pose a challenge for the management of stocks shared across spatial regulatory boundaries [10, 12–14]. When a stock falls within two or more spatial regulatory units, managers must often coordinate efforts to maintain sustainability of the (shared) stock [15]. However, there can be a growing disconnect between the region of the ocean where fish are found and the management jurisdictions which hold harvest rights [16]. The requirement for clearly defined resource boundaries is a fundamental principle of managing a common pool natural resource where market failures such as "Tragedy of the Commons"-type negative externalities and "Free Rider"-type positive externalities exist across jurisdictions. This is especially the case when the resource is not managed as a whole but is instead partitioned into multiple jurisdictions. Such market failures generally result in inefficient resource utilization [15]. The mismatch between implemented management strategies and shifts in stocks across management jurisdictions compromises management and sustainability efforts [10–14, 17]. Shifts of important fish stocks have resulted in both historically unsustainable harvest and international conflict as in the case of the United States (U.S.) and Canada over sockeye salmon (*Oncorhynchus nerka*) and the European Union and Iceland over Atlantic mackerel (*Scomber scombrus*) [13, 18].

Along the U.S. Northeast Atlantic coast, many economically important fished species such as American lobster (*Homarus americanus*), yellowtail flounder (*Limanda ferruginea*), summer flounder (*Paralichthys dentatus*) and red hake (*Urophycis chuss*) have been shifting their distribution poleward [17]. Modeling exercises suggest that under a high emission (i.e., extreme) climate change scenario, species in the region could shift an average of more than 600 km by the end of the 21$^{st}$ century [19]. Fishers in the U.S. Northeast Atlantic have been adapting to such shifts by following the fish, in some cases, or changing target species, in others [20]. Recent research looking at mismatches between spatial regulatory tools and climate change driven shifts in species distributions have called for a transition to dynamic management rules [8, 10, 17]. However, this transition is particularly difficult for two main reasons: (*i*) management jurisdictions in the ocean are often ill-defined, especially at the sub-national level or beyond Exclusive Economic Zones (EEZs), and (*ii*) quantifying fish distributions in relation to spatial regulatory boundaries is difficult, even where fishery-independent surveys exist, because of seasonal and interannual variation.

Within U.S. Atlantic waters, each state allocates fishing rights within its own state waters (i.e., a defined regulatory unit of about 3 nm. from land where states have authority to

distribute fisheries rights). However, federal waters (i.e., the 200 nm, EEZ) are not subdivided into sub-regulatory units, except for some specific cases, and in most cases, fishing occurs without state-specific zoning. Nevertheless, a challenge for all fixed spatial regulatory units used for allocating fishing rights in the ocean is that fish distributions may change with seasonality, which is a key consideration in temperate fisheries management [21]. Accounting for seasonality in surveys allows us to understand spatial patterns of reproduction, seasonal phenology and long-term dynamics of a stock [22, 23]. Along the U.S. Northeast EEZ, many species such as black sea bass (*Centropristis striata*) and scup (*Stenotomus chrysops*) migrate latitudinally and onshore/offshore in response to seasonal changes in water temperature [24, 25]. Longer term changes in depth distribution can also occur, as in the case of summer flounder (*Paralichthys dentatus*) which undergo annual migrations from shallow coastal habitats in the fall to deeper habitats over winter [26, 27]. The consideration of such seasonal migration, in conjunction with migration driven by a changing climate, is key to the proper management of marine resources [21, 23].

Here, we examine the challenge of designing spatial boundaries for the allocation of shifting fish stocks within EEZs. We do this by quantifying the historical distributions of three important fish stocks relative to 11 Northeast U.S. states that share management of these stocks with each other and with the Mid-Atlantic Fishery Management Council (MAFMC). Specifically, we ask how the choice of survey timing and regulatory unit design influences our perception of fish biomass distribution changes relative to management boundaries. To answer this question, we first determine the time series of changes in the historical distribution of the three stocks along the U.S. Northeast EEZ. Second, we follow two alternative approaches to designing regulatory units. Finally, we consider two seasons to allocate the distribution of fish stocks across the 11 states. We discuss the implications of our results for the broader challenge of management of shared stocks across ill-defined management boundaries.

## Methods

### Fish stock selection

The current analysis focused on black sea bass, summer flounder, and scup, stocks managed jointly by the MAFMC and the Atlantic States Marine Fisheries Commission (ASMFC). The MAFMC and the ASMFC work cooperatively to develop commercial and recreational fishery regulations for summer flounder, scup, and black sea bass. The management unit extends from the U.S.-Canadian border to the southern border of North Carolina for summer flounder, and Cape Hatteras (North Carolina) for scup and black sea bass (MAFMC: https://www.mafmc.org/sf-s-bsb/) (Fig 1).

These fisheries are managed cooperatively in state waters (0–3 nautical miles) by the MAFMC, the ASMFC and the individual states. It is worth mentioning that while the annual coast-wide quota is allocated among states, and each state allocates quota within its own state waters, fish found in the U.S. Federal waters are not allocated to particular states. For Federal waters fisheries, an annual coast-wide commercial quota, based on historical landings, was implemented, and since 2003 further allocated to the states of Maine, New Hampshire, Massachusetts, Rhode Island, Connecticut, New York, New Jersey, Delaware, Maryland, Virginia and North Carolina [28] (Table 1). While this work focuses on the consequences of stock shifts to their management within Federal waters, changing fish distributions will also affect fisheries in state waters. For summer flounder, the historical period on which the commercial quota allocations are based is 1980 to 1986, as determined in the original management plan [29]. For scup, the period covers 1988 to 1992 according to Amendment 8 of the original management plan [30], and for black sea bass, Amendment 13 sets the state-specific commercial quota

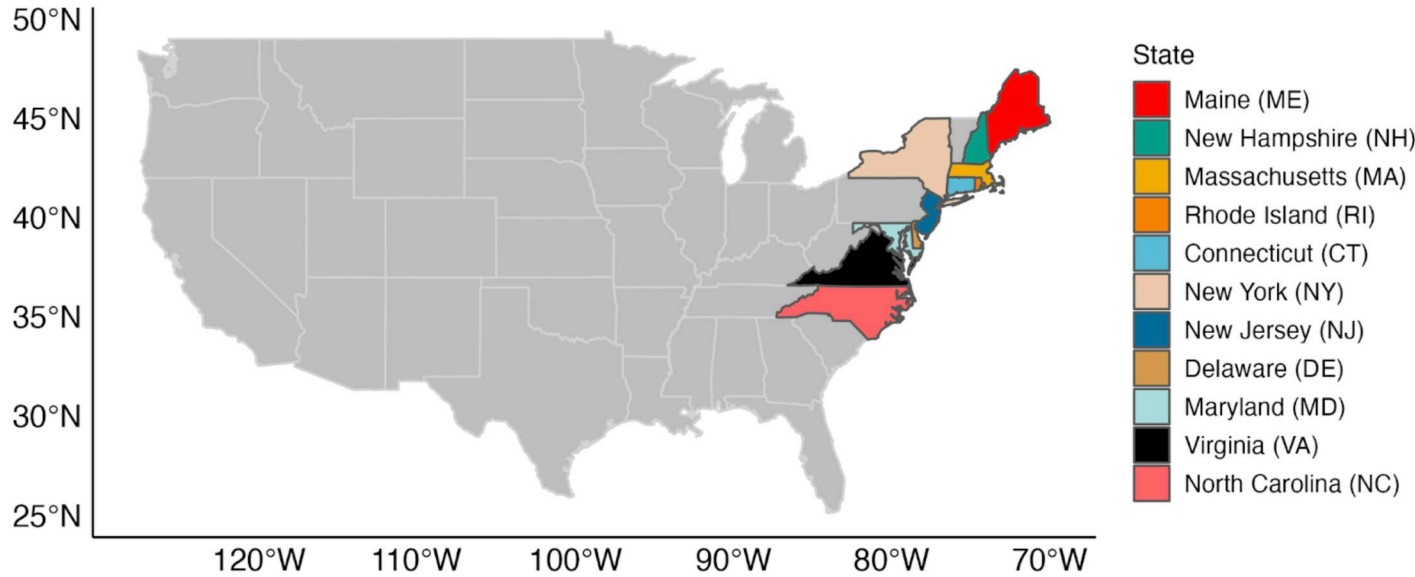

**Fig 1. Location of the states included in this study.** States presented in legend in latitudinal order.

based on historical landings from 1980 to 2001 [28]. These time periods were adapted here as reference points for each stock. It is worth mentioning that the recreational fishing sector plays an important role in fishing effort for these stocks [31]. However, recreational harvest is not tracked in the same way as commercial harvest because it is not landed at licensed dealers. Thus, it was not included in our analysis. Future research should aim to incorporate this sector for a better understanding of fishing pressure on these shifting stocks.

## Data used

Distribution data for all three stocks were gathered from NOAA's yearly Northeast Fisheries Science Center Spring and Fall Bottom Trawl Surveys accessed through the *Ocean Adapt* portal (https://oceanadapt.rutgers.edu) and standardized as catch per unit of effort (CPUE) defined as trawl-catch in weight per area swept (kg/ha) [32]. Commercial fishery landings data

**Table 1. State-wide commercial quota allocation for black sea bass (*Centropristis striata*), summer flounder (*Paralichthys dentatus*) and scup (*Stenotomus chrysops*) according to the Mid-Atlantic Fishery Management Council.** Values shown as percentage of total quota. States ordered latitudinally from north to south. Note that summer flounder quota for Maine, New Hampshire and Delaware is 0.05, 0.0005, and 0.02 percent, respectively.

| State | Black sea bass (*Centropristis striata*) | Summer flounder (*Paralichthys dentatus*) | Scup (*Stenotomus chrysops*) |
|---|---|---|---|
| Maine | 0.4 | 0 | 0.1 |
| New Hampshire | 0.4 | 0 | 0 |
| Massachusetts | 15.6 | 6.8 | 21.6 |
| Rhode Island | 13.2 | 15.7 | 56.2 |
| Connecticut | 3.7 | 2.3 | 3.2 |
| New York | 8.6 | 7.6 | 15.8 |
| New Jersey | 20.1 | 16.7 | 2.9 |
| Delaware | 4.1 | 0 | 0 |
| Maryland | 8.9 | 2 | 0 |
| Virginia | 16.1 | 21.3 | 0.2 |
| North Carolina | 8.9 | 27.4 | 0 |

by port and state were provided by the Atlantic Coastal Cooperative Statistics Program (ACCSP https://www.accsp.org) [33]. We used only publicly available data that were identified at the port level. This comprised 45% of the total available landings; the remaining 55% was private data or was not identified at the port level. Landing ports for North Carolina black sea bass were unknown for all years of data. In this unique case, we attributed all NC landings of this species to the port of Wanchese, the largest port in NC in terms of landings [31]. The present analysis focuses on the U.S. Atlantic EEZ using the *Sea Around Us* shape file (updated 2015; http://www.seaaroundus.org) and the Federal and State Waters shapefile created for the Ocean Reporting Tool [34]. This shapefile consists of multiple polygons each one representing the statutory 3-mile limit of each state's marine waters.

## Fish stock interpolation

We interpolated the trawl survey data from the observational points to the U.S. EEZ from the southern border of North Carolina to the U.S.-Canadian border for both Fall and Spring seasons covering the time period of 1971 to 2019. The EEZ shapefile was gridded to a 0.3˚ x 0.3˚ matrix and trawl survey CPUE was interpolated with a triangular irregular surface (TIS) method using the *R* [35] package *interp* [36]. Overall, the TIS method creates multiple non-overlapping triangles that cover the entire interpolated region (i.e., the EEZ) from an initial set of sample points (i.e., NOAA's trawl survey sampling sites) generating a spatially-continuous surface [37]. The interpolation was done separately for each season and year.

## Determining regulatory units

Although the stock-wide quota for our three target stocks is allocated among states, U.S. Federal waters are not sub-divided into state-owned fisheries management regions (hereafter referred to as regulatory units). This lack of subdivision of federal waters into smaller regulatory units creates a challenge for assessing a stock's distribution with respect to individual states. There are any number of ways to geographically determine regulatory units within the federal waters of the EEZ. Here we focus on two methods which likely bracket much of the range of options.

**i) State waters expansion.** The first approach consists of expanding the current geographical distribution of statutory state waters (Fig 2– *State waters*). For this approach we set the state waters shapefile from Maine to Virginia as the starting point (Fig 2– Step one). We then expanded each state's boundary out to the 200 nm EEZ using the *st* package [38] in *R* (Fig 2– Step two). Each extended polygon was then cropped to the shape of the EEZ (Fig 2– Step three). Finally, the resulting polygons (i.e., regulatory units) were gridded into a 0.3˚ x 0.3˚ matrix. The regions where the grid-cells overlapped were equally allocated to the sharing states.

**ii) Fishing ports approach.** The second approach to determine regulatory units gives greater weight to each fishing port's landings (Fig 2– *Fishing ports*). First, we used landings from the ACCSP data for each stock and port averaging them from 2010 to 2019. We selected these years as they are the most recent years of data available and because they are the furthest away from when the quota was established. However, we recognize that the years of landings data selected might influence the results, and the results should be interpreted with this caveat in mind. We then estimated the total port-wide landings of the averaged period and the proportion attributable to each port of landings *per* stock. We selected the main ports landing each stock by sequentially selecting the top 25% of ports according to the weight of landings for each stock. Using black sea bass as an example, out of a total of 76 ports with commercial landings, we selected the 19 ports with the largest landings by weight (i.e., (19/76) * 100% =

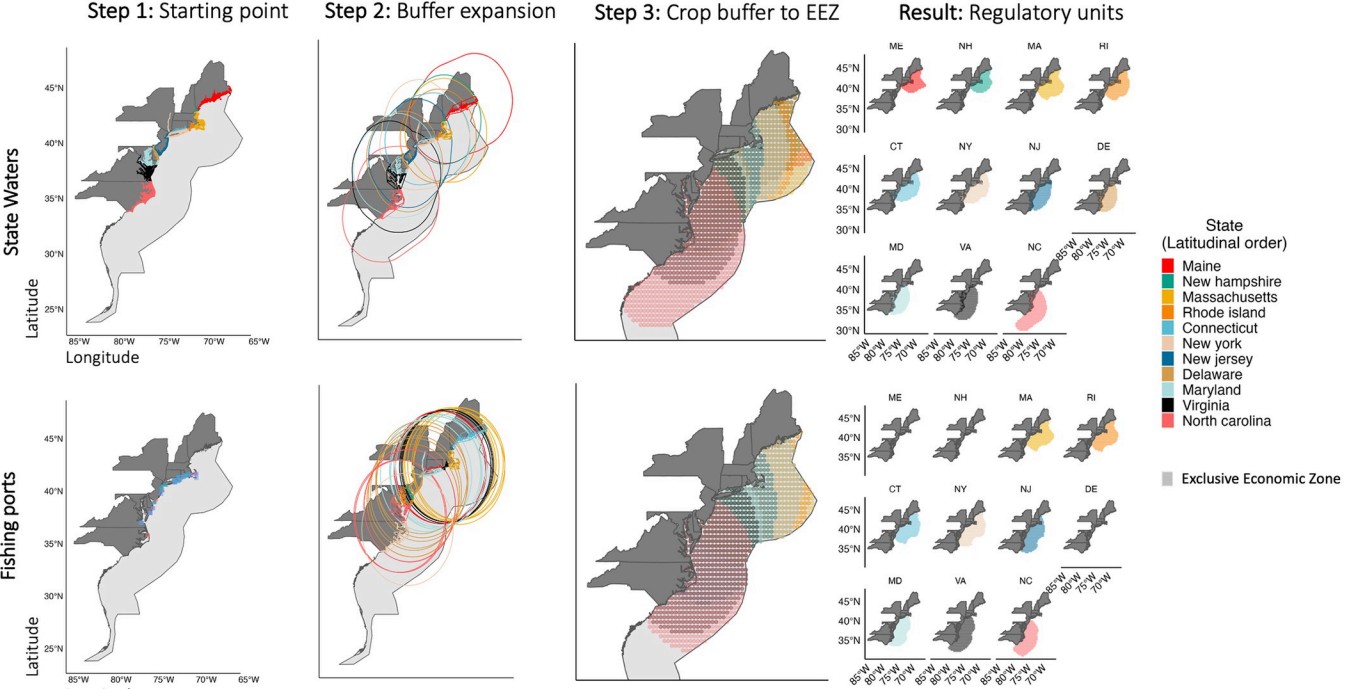

**Fig 2. Graphical description of the method employed to design the regulatory units.** Top row represents the state waters approach. Bottom row represents the fishing ports approach. Both approaches follow the same methods according to each column title. Colors represent each state's expanded regulatory unit in each time step. Step one is the starting point (state waters top; fishing ports bottom), step two shows the expanded buffer from the starting point, step three shows that same buffer cropped to the EEZ (note that the area of each state's cropped buffer is shaded), the result shows the final regulatory unit by state (similar to step 3 but not-overlapped).

25% of the ports). This resulted in 19 ports for black sea bass representing 93% of landings, 16 for scup representing 95% of landings, and 23 for flounder representing 97% of landings. Note that, while stocks often shared top ports, the final number and location of ports varied according to each stock. Each port is represented by a point on the map (i.e., latitude and longitude degrees) whose coordinates were manually collected using GoogleMaps (https://www.google.com/maps) (Fig 2– Step one). We then expanded a buffer of a given radial distance from each port to the EEZ (Fig 2– Step two) and cropped the resulting buffer to the EEZ (Fig 2– Step three) following the same method as described above for the State waters approach. Just like the previous approach, the final regulatory units were also gridded. The resulting regulatory unit of each port was associated with the state where the port was located (Fig 2). Overlapping grids were equally allocated to the sharing states. We acknowledge that our approaches are based on GIS methods of buffering and that designing regulatory units based on other methods such as historical use patterns by port [39] or fishing sites [40, 41] would be more management-relevant.

We tested the sensitivity of our results to the arbitrary 25% threshold and found that the threshold did not influence the result as much as having or not a port landing the stock. That is, the number of grid-cells that each state had by percentile did not vary as much as not having a port at all (S1 Fig).

## Summarizing fish stock distributions with respect to regulatory units

Once regulatory units were drawn the next step was determining the proportion of each fish stock's distribution falling within each state's regulatory unit in each year. For each stock, we

first overlaid the yearly trawl survey grids with the states' regulatory units (i.e., state waters expansion and fishing ports approach). Second, we calculated the yearly proportion of the distribution that each state held within its regulatory unit. We did this by first aggregating the trawl survey data of all grid-cells that fell within a state's regulatory unit and then dividing the sum by the coast-wide trawl survey data. In cases where two or more states had the same grid-cell, the trawl survey data for the overlapping unit was divided equally among the states. For example, if New York and New Jersey both shared one grid-cell, then the proportion of the distribution falling within that grid cell would be allocated 50% to each state. Alternatively, if Connecticut also shared the same grid cell, then the proportion would be 33.3% to each state. Note that this step did not affect the trawl survey interpolation method but did affect the way the resulting stock spatial distribution was allocated to each state.

Once we had a year-state estimate of each stock's distribution proportion between 1971 and 2019, we averaged the stock distribution proportion in 2010–2019 for each state. We then compared the distribution proportion of the reference time period of each stock (the time period of historical catches that was used to actually allocate harvest rights) to the 2010–2019 period. The average of 2010 to 2019 was selected to represent the more recent data available while the reference time period was determined by the years used by the MAFMC to set the state-wide quota for each stock (see Fish *stock selection*). The percent change between the reference and 2010–2019 time periods in the distribution of the stock ($\Delta\alpha$) was estimated per state and per stock as follows:

$$\Delta\alpha = \frac{(\alpha_{current} - \alpha_{reference})}{|\alpha_{reference}|} * 100 \qquad (1)$$

Where $\alpha_{current}$ is the average proportion of the stock distribution falling within a state's regulatory unit for the period from 2010 to 2019 and $\alpha_{reference}$ is the average for the reference time period.

We estimated $\Delta\alpha$ using separate regulatory units for fishing ports ($\Delta\alpha_{fp}$) and state waters ($\Delta\alpha_{sw}$). In each case, we first used the fall ($\Delta\alpha_f$) and then the spring ($\Delta\alpha_s$) interpolation. Ultimately, we calculated the differences in $\Delta\alpha$ between survey seasons ($\Delta\alpha_{seas}$) and between regulatory unit types ($\Delta\alpha_{reg}$) approach by subtracting from each other:

$$\Delta\alpha_{reg} = \Delta\alpha_{fp} - \Delta\alpha_{sw} \qquad (2)$$

$$\Delta\alpha_{seas} = \Delta\alpha_f - \Delta\alpha_s \qquad (3)$$

Note that $\Delta\alpha_{reg}$ was estimated using the spring interpolation. All of the analysis was implemented using *R* version 4.1.0 (*Camp Pontanezen*) and associated packages (S1 Table). All code can be found in https://github.com/jepa/AcrossBoundaries and an interactive online data exploratory tool of the results can be found at http://jepa.shinyapps.io/allocation_tool/ (See data availability statement for guidance on data availability).

## Results

The choice of boundary expansion method (e.g., fishing port radius or extension of state boundaries) is highly consequential for determining the spatial extent of each state's regulatory unit (Fig 2). For example, using the state boundaries approach, Delaware and Maine have regulatory unit areas of 259 and 213 km$^2$, respectively. Because neither state has a port with commercial landings of black sea bass, scup, or summer flounder within the top 25% of landings, their regulatory units based on the fishing ports approach include only their state waters out to

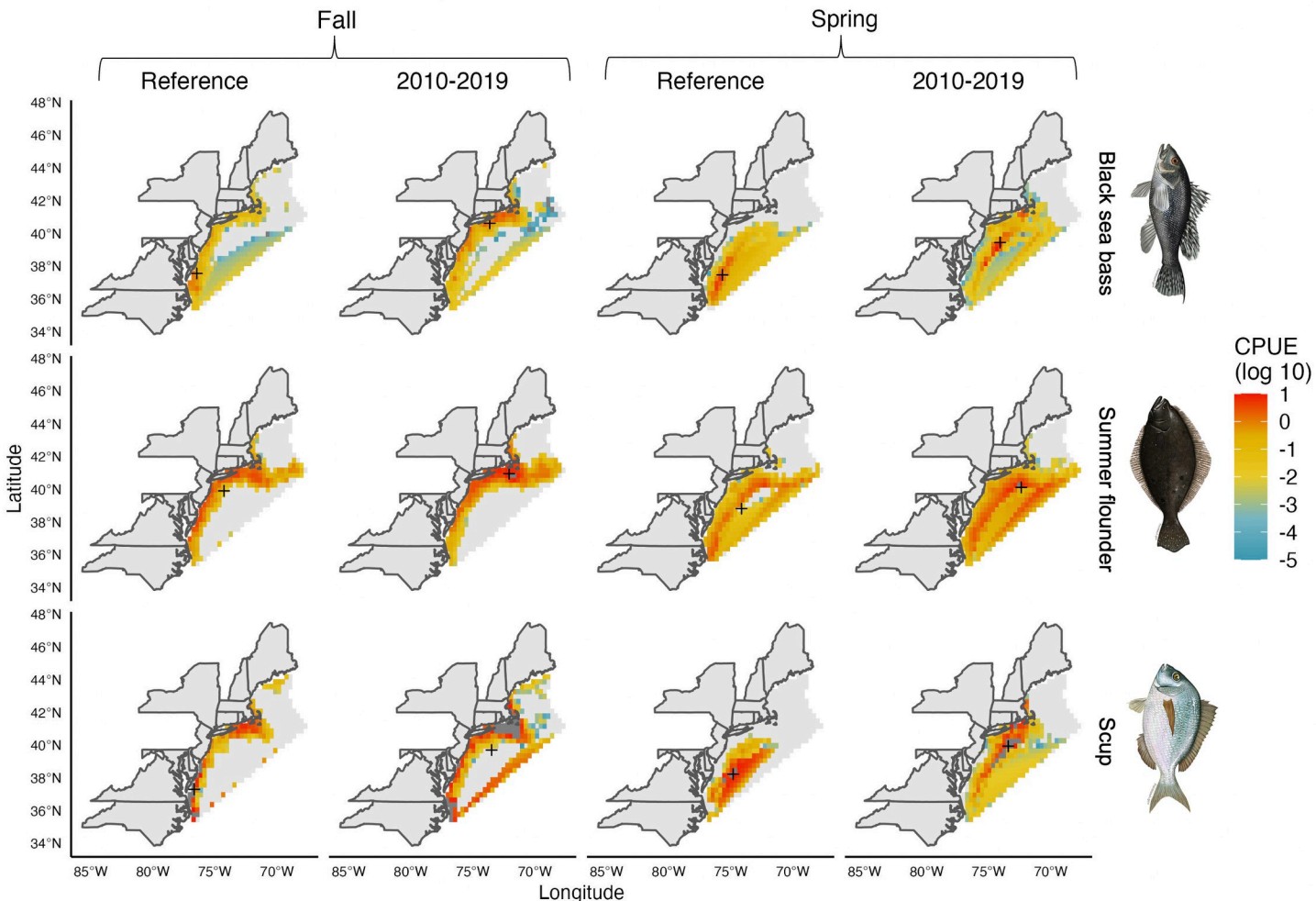

**Fig 3. Interpolation results for all species and seasons within the US Northeast Exclusive Economic Zone.** Scale showing the trawl survey data as Catch per Unit of Effort (CPUE) at a log 10 scale. Black cross represents the species' region with the largest CPUE with grey regions representing no CPUE. Reference period is related to the quota allocation period for each stock (see methods). Fish images from MAFMC (www.mafmc.org/sf-s-bsb).

3 nm, reducing these areas. In contrast, New York's regulatory unit is 24% larger under the regulatory unit drawn under the fishing port approach because New York has the greatest number of ports within the 25% of landings (*n* = 7) and the greatest distance between ports (~145 km separating Montauk and Point Lookout).

Seasonality plays an important role in determining the stock distribution of black sea bass, scup, and summer flounder (Fig 3). Latitudinal variations in distributions are notable for scup, whose historical northern limit goes from Massachusetts (latitude ~42.4˚) in spring to the northern coast of Maine (latitude ~44˚) in the fall. Moreover, both scup and black sea bass have expanded their distributions northward. For example, the spring Northern limit of black sea bass expanded from New Jersey (latitude ~40.6˚) in the late 1980s to the border of Maine and New Hampshire (latitude ~43˚) in 2019. In addition, the region with the highest trawl survey data for sea bass has shifted 3˚ north in the fall since 1980 (2˚ in spring), which is the greatest shift observed among these three species. In the case of summer flounder, however, such latitudinal seasonality is less evident (1˚ north for both seasons). Instead, the interpolation results suggest a longitudinal (or depth) expansion in the species range from a largely coastal

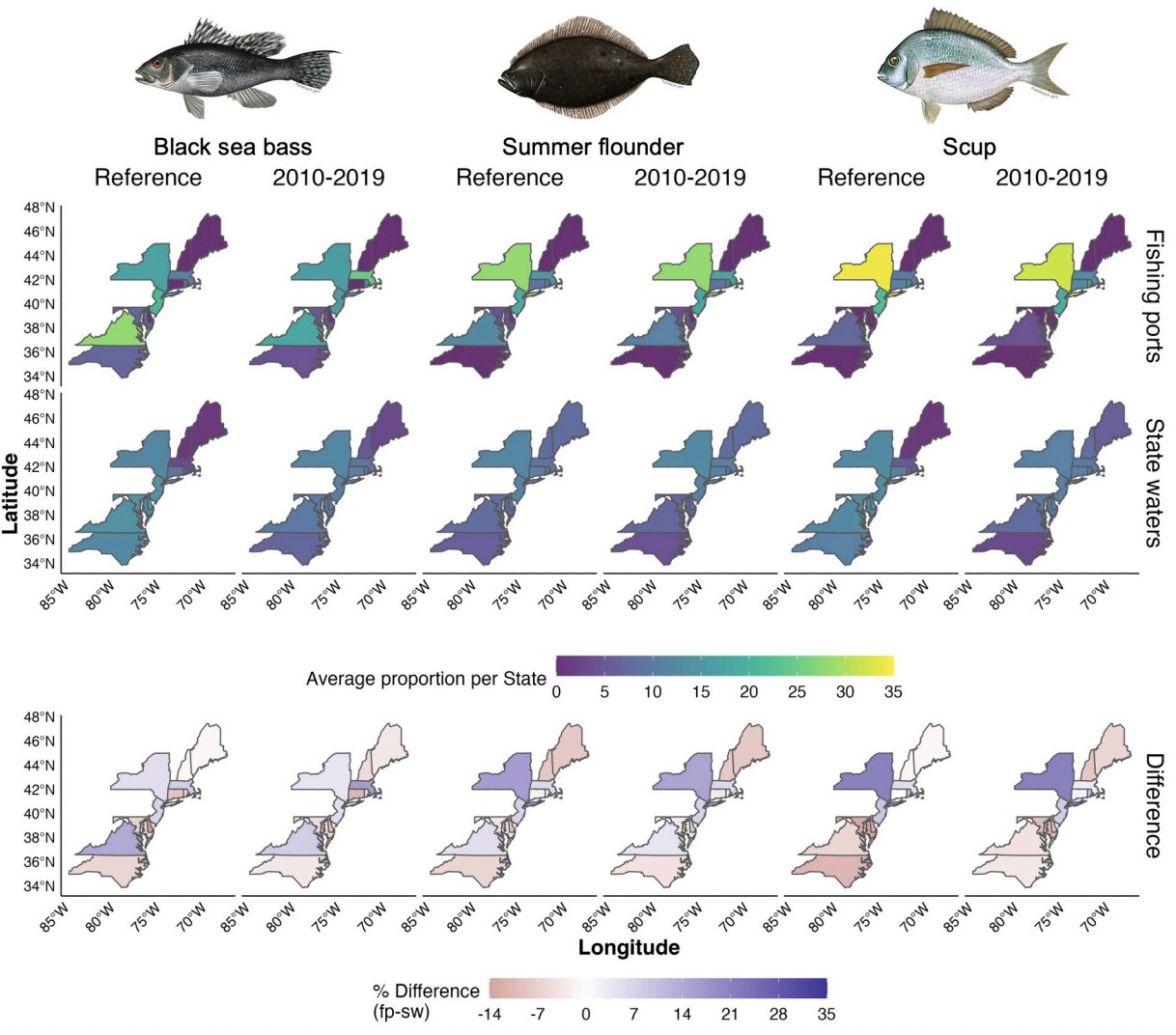

**Fig 4. Proportion of trawl survey data as Catch per Unit of Effort (CPUE) for each state and species by spatial approach.** Top and middle rows represent the fishing port and state waters approach, respectively. Bottom row is the difference between both approaches ($\Delta\alpha_{reg}$; see methods). All values in percentage units. Fish images from MAFMC (www.mafmc.org/sf-s-bsb).

concentration in the fall to an eastward (offshore) shift in the spring. Such a shift in the spring season seems to have become even more pronounced in recent years relative to historical levels (Fig 3). While the geographical characteristics of the coast pushes all stocks eastward as you move north, scup seems to be the only stock that is shifting both latitudinally and further off-shore with a northward shift of 2˚ in both seasons and a longitudinal shift of 3˚ in the fall.

We overlaid the putative regulatory units with the interpolated annual stock distributions of black sea bass, summer flounder and scup to estimate each state's proportion of the total stock distribution (Fig 4). Under the state waters approach, each state has a regulatory unit and thus captures a part of the stock distribution of each species. However, under the fishing ports approach some states end up without regulatory units and thus the proportion of the stock distribution attributed to these states was zero in all years. This is the case of Maine, New Hampshire, and Connecticut. In contrast, New Jersey and Virginia end up with the largest benefits from the fishing ports approach. There are very few cases where the choice of

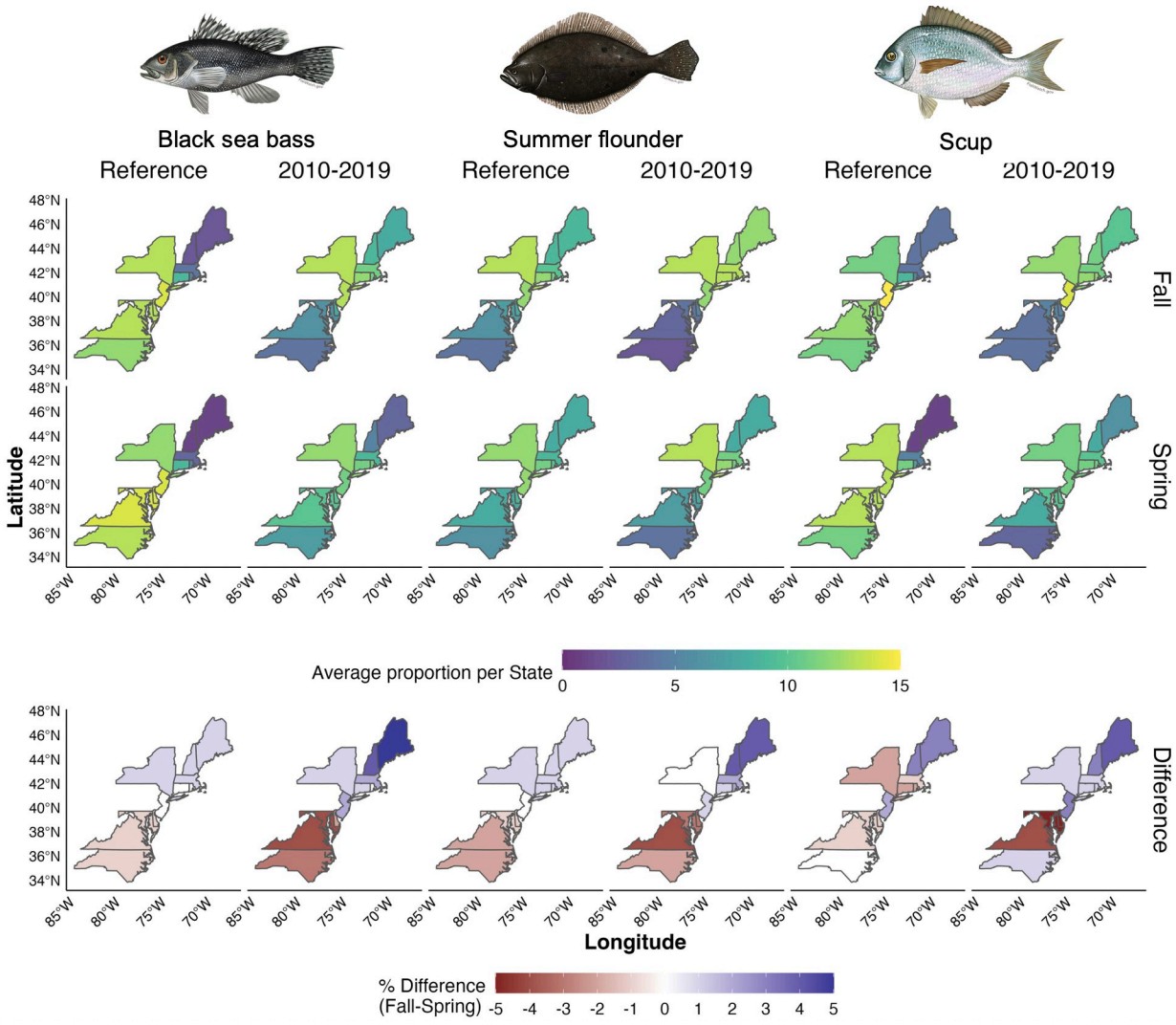

**Fig 5. Proportion of trawl survey data as Catch Per Unit of Effort (CPUE) for each state and species by survey season.** Top and middle rows represent fall and spring surveys, respectively. Bottom row is the difference between both approaches ($\Delta\alpha_{seas}$; see methods). All values in percentage units. Fish images from MAFMC (www.mafmc.org/sf-s-bsb).

regulatory unit approach does not substantially affect the distribution proportion of each state (Delaware, Massachusetts and North Carolina for black sea bass, North Carolina for summer flounder, and Maryland for scup).

Seasonal differences in the proportion of the stock distribution falling within each state's regulatory unit are also evident (Fig 5). Here, half of the cases represent 1% or no seasonality difference like Connecticut's distribution proportion of black sea bass and summer flounder in the reference period. The largest seasonal differences are in the latitudinal extremes. In recent years, Maine has experienced a 5% increase in fall distribution proportion, compared to spring, whereas Virginia experienced the opposite: 5% higher distribution in spring than in fall.

Over the last 40 years, the proportion of the distribution in each state has increased for northern states and decreased for southern states for all three species (Fig 6). North Carolina and Virginia are the most affected, regardless of the approach used to establish putative

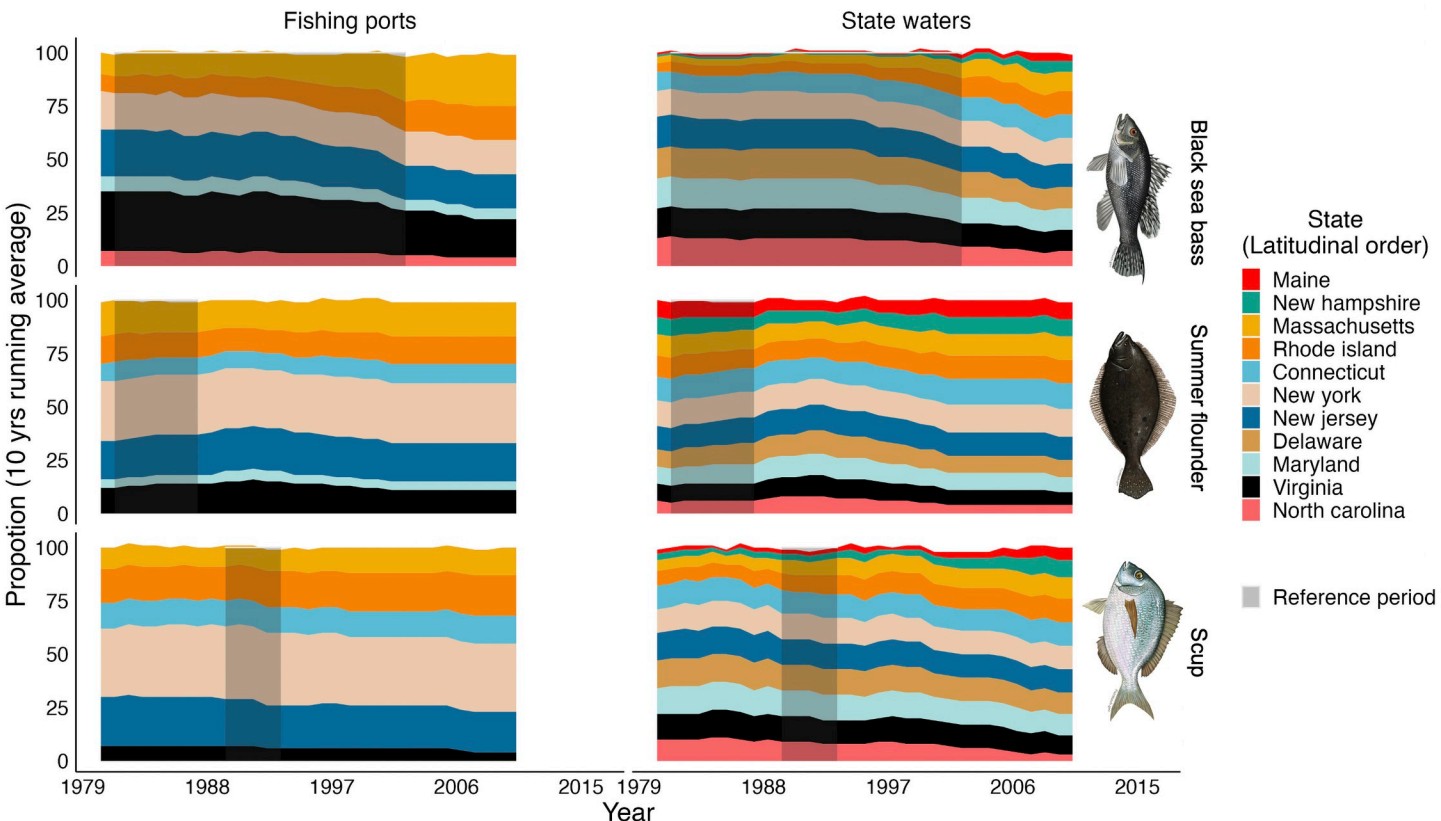

**Fig 6. Historical proportion of trawl survey data as Catch Per Unit of Effort (CPUE) for each state and species by spatial approach.** Proportion presented in 10 years running mean percentage. States presented latitudinally from north to south. Grey shade represents the reference period for quota allocation. Fish images from MAFMC (www.mafmc.org/sf-s-bsb).

regulatory units. Under the state waters approach, states from Rhode Island to Maine increased their proportion of black sea bass and scup while the fishing ports approach favors Massachusetts, Rhode Island, and New York. The proportion of the distribution for states located at "middle latitudes" (e.g., New York, and Connecticut) does not seem to be affected by shifting distributions. While the fishing ports approach to designing regulatory units can reduce the distribution proportion of some states to zero (e.g., Maine), other states benefit strongly (e.g., New Jersey). While the extent of the change in distribution proportion to each state varies between approaches, the overall 'winners' and 'losers' are the same, regardless of the approach.

## Discussion

Marine species have been shifting globally in response to climate change [1, 32] and are expected to continue doing so at least through the end of the century [4, 5]. Such shifts challenge ocean governance, particularly when dealing with shared stocks that cross international and sub-national management boundaries [10, 12, 14, 17]. Fisheries managers are having to design new management strategies. This is the case for the MAFMC and the ASMFC, which jointly manage black sea bass, summer flounder, and scup along the Northeast Atlantic coast of the U.S. However, as our results show, designing these new strategies is not straightforward, as numerous technical decisions (e.g., regarding boundary definitions or which survey to use when defining stock distributions) can have important implications.

Spatial boundaries of ocean management tools can be drawn based on different approaches such as distance to fishing ports, community historical usage of a certain resource, stakeholder engagement or optimization algorithms. For example, in many small-scale fisheries across Latin America spatial property rights are often given to fishing communities based on historical exploitation and investment in specific fisheries [40, 41]. In Chile, the government adopted the use of Territorial Use Rights for Fisheries (TURFs) for the management of benthic fisheries. Under this policy, fishers' organizations are allocated a TURF based on port location and historical landings, among other considerations [42, 43]. Moreover, optimization algorithms such as MARXAN [44] are a popular tool for designing areas protected from human activities with examples in the United Kingdom [45], the Great Barrier Reef [46], and the Caribbean-wide marine protected areas [47]. Optimization approaches allow for the integration of data to maximize conservation objectives while considering social costs to deliver a series of "optimal" solutions [44, 45].

Similarly, the joint management of shared stocks relies on defined boundaries (i.e., regulatory units) where fishing rights are allocated to share-holders [15]. In the present case of shared stocks in the Northeast US, the method for designing regulatory units in federal waters could have large consequences for the allocation of quota to each state (Figs 3–5). The fishing ports approach relates to many cases around the world, where spatial fishing rights are allocated to share-holders based on historical landings [11, 42] and proximity to fishing grounds [40, 41]. These methods are often considered fair in that they respect the historical investment of individuals in the fishery. However, such an approach has the potential to exclude some users from the fishery, particularly in the case of small scale fisheries [48], as in the cases of Maine, New Hampshire, and Connecticut for some scenarios considered here (Fig 2). Moreover, because this approach is based on historical landings, it does not necessarily account for future biomass changes across regulatory units, nor expansion to new regulatory units. With shifting fish stocks, an approach that results in the exclusion of either historic or potential new share-holders can result in conflict or unsustainable harvest as seen in multiple fisheries across the world and within the U.S. [14, 18, 42].

Designing regulatory units based on an expansion of the state water boundaries into the federal EEZ would not exclude shareholders with historically small landings. In our study, the state waters expansion, by giving regulatory units to Maine, Connecticut, and New Hampshire, accounts for the shifts in stock biomass and allows these states to benefit from an emerging local fishery as stocks move poleward (Figs 3 and 5). Allocating quota to individuals who have not formerly participated in a fishery might not be seen as fair to those shareholders who have historically invested in and relied on the fishery. From a legal perspective, for federally managed fisheries in the U.S., including the three stocks in our analysis, allocation of fishing privileges needs to follow National Standard 4, which requires all such actions to be fair and equitable, as well as National Standard 8, which requires that management measures provide for the sustained participation of fishing communities and, to the extent practicable, minimize adverse economic impacts on these communities [49].

Determining what is "fair and equitable" requires an understanding of the reliance of different regional groups on the fishery, as well as their capacity to adapt to changes in stock distributions and quota allocation. Fishers, particularly those with large and mobile vessels, might be willing and able to navigate longer distances to "follow the fish" rather than changing to target more locally abundant species or dropping out of the fishery [20]. Additionally, the ability of a fisher to change targeted species depends on the regional abundance of alternative species (which may also be undergoing similar distributional shifts), access to permits or quota for those species (which may be expensive or unavailable), markets, and many other factors. In

some cases, like the Philippines, the design of fisheries management units was based on the distribution of stocks, structure of fisheries and administrative divisions [50].

Allocating quota based on current species distributions, rather than historical reference points, has the potential to improve the resilience of management plans to climate change driven shifts in stock distribution. In the northeast Pacific, for example, the joint management of Pacific halibut (*Hippoglossus stenolepis*) by Canada and the U.S. relies on regulatory units that extend along the countries' EEZs from California to Alaska [51]. Modeling exercises suggest that the management plan is resilient to climate change as quota is allocated to each regulatory unit based on the current distribution of the stock [52, 53]. A combination of both historic and current distributions has been used to manage other shared stocks. For example, the Transboundary Resources Assessment Committee in the Gulf of Maine recommends quotas for cod (*Gadus morhua*), haddock (*Melanogrammus aeglefinus*), and yellowtail flounder (*Limanda ferruginea*) by employing a weighted algorithm that assigns 90% of the weight to a stock's current distribution and 10% to its historic distribution [54]. Here, we found differences in the distributions of the stocks between the years when the quotas were determined (i.e., historical period) and current values (Figs 2 and 3). These results suggest that current allocation policies might not be efficient in terms of allocating biomass to states, especially for black sea bass and scup. Recent revisions to the commercial state allocations for black sea bass and summer flounder have recognized shifting distributions [55, 56], but are hampered by the lack of an agreed set of regulatory units at the state level and a procedure for defining and allocating the stock distribution to these regulatory units. Our results provide one set of proposals for how such a dynamic spatial allocation mechanism could be developed.

In general, a transition to dynamic allocation will likely face some resistance from fishers who have historically fished the stock. Fisheries permits often move from open access to limited access as fishing pressures on the stock increase. If limited access results in a larger fish stock, limited access permits tend to gain value, reflecting the discounted expected benefits of larger future landings [57]. However, if fish stocks shift outside the management region, limited access permits may lose value (due to either the reduced profitability of the longer fishing trips required to access the more distant fish stock, or the costs of moving fishing operations to a different management region, which would reduce the price in-region fishers would be willing or able to pay for a permit). This could be mitigated by allowing fishers to sell or lease permits to fishers in a different management region with access to the relocated fish stock, however transferability is often controversial. Permit holders may also engage in fishing practices that are short term profitable but harm the long-term sustainability of the stock, exacerbating "Tragedy of the Commons" situations across jurisdictional boundaries. For example, fishers in jurisdictions with emigrating stocks may reduce conservation efforts, as fishers in other jurisdictions will likely receive most of the benefits. Limited access permit owners who are prohibited from transferring their permits might reasonably expect compensation for any loss of permit value due to increased fishing trip costs or moving costs. In contrast, limited access permit owners in regions with in-migrating stocks would likely see an increase in permit value, reflecting the discounted expected benefits of larger current and future landings—in part due to the stewardship activities of fishers in the trailing part of the range distribution shift. In theory, the increases in permit value in regions with in-migrating stocks could be used to compensate for the decreases in permit value in regions without migrating stocks. Whether there would be a net gain to fishers across all regions combined would depend on whether the fish move to a region where catching them is more profitable, or to a region where catching them is less profitable, compared to the profitability of catching them in their region of origin.

## Conclusion

Adaptive regulations that can adjust to current species distributions have the potential to provide a more efficient and potentially more sustainable solution to managing fisheries impacted by climate change. However, here we demonstrate that the methods used to measure current distributions and determine regulatory unit boundaries will affect how fishing rights are allocated across different parties. We provide two possible options, one based on existing fishing activity (i.e., based on landing ports) and the other on potential access to resources (i.e., based on projected state boundaries), along with considerations for guiding that choice, such as socioeconomic impacts on existing practices, the carbon footprint of pursuing shifting fishing grounds, marketability of newly available species, and enabling legal frameworks. Each approach results indifferent "winners" and "losers", and it will be important to consider potential compensation systems to work toward more equitable distribution of socioeconomic benefits and impacts. To select which allocation approach to apply, i.e., whether favoring historical rights, new opportunities and/or inclusivity of minority rights-holders, as well as deciding if an adaptive approach should be introduced at all, it will be key to consider the broader context of social, environmental, and economic change associated with multiple linked fisheries and the industries reliant on, or contributing to, those fisheries.

## Supporting information

**S1 Fig. Number of grid cells that each state would have under the ports approach according to the percentage of top fishing ports selected.** Each color represents the top percentage of ports relative to total landings. This study used the top 25% of ports that landed the greatest number of fish for each stock (In purple and bold in legend).
(PNG)

**S1 Table. List of main R packages used in this project.**
(CSV)

## Author Contributions

**Conceptualization:** Scott Crosson, Rod Fujita, Arielle Levine, Olaf P. Jensen.

**Formal analysis:** Juliano Palacios-Abrantes.

**Funding acquisition:** Scott Crosson, Rod Fujita, Arielle Levine, Olaf P. Jensen.

**Methodology:** Juliano Palacios-Abrantes, Rod Fujita, Arielle Levine, Catherine Longo, Olaf P. Jensen.

**Project administration:** Arielle Levine, Olaf P. Jensen.

**Supervision:** Olaf P. Jensen.

**Visualization:** Juliano Palacios-Abrantes, Catherine Longo, Olaf P. Jensen.

**Writing – original draft:** Juliano Palacios-Abrantes, Scott Crosson, Chris Dumas, Rod Fujita, Arielle Levine, Catherine Longo.

**Writing – review & editing:** Juliano Palacios-Abrantes, Scott Crosson, Chris Dumas, Rod Fujita, Arielle Levine, Catherine Longo, Olaf P. Jensen.

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
