## [Decision Letter · Decision Letter 0]

29 Aug 2022

PONE-D-22-11430Quantifying fish range shifts across poorly defined management boundariesPLOS ONE

Dear Dr. Palacios-Abrantes,

Thank you for submitting your manuscript to PLOS ONE. After careful consideration, we feel that it has merit but does not fully meet PLOS ONE’s publication criteria as it currently stands. Therefore, we invite you to submit a revised version of the manuscript that addresses the points raised during the review process.

Two reviewers have now commented on the manuscript and raise several concerns regarding the methodology and interpretation of the results, which need to be addressed.  They also make additional suggestions to improve the overall framing in the introduction, description of methods, and clarity of presentation that have to be taken into account for a revised version.  For details, please refer to the reviewer comments below.

We look forward to receiving your revised manuscript.

Kind regards,

Caroline Ummenhofer

Academic Editor

PLOS ONE

Journal Requirements:

Reviewers' comments:

Reviewer's Responses to Questions

**Comments to the Author**

1. Is the manuscript technically sound, and do the data support the conclusions?

Reviewer #1: Yes

Reviewer #2: No

2. Has the statistical analysis been performed appropriately and rigorously? 

Reviewer #1: Yes

Reviewer #2: No

3. Have the authors made all data underlying the findings in their manuscript fully available?

Reviewer #1: Yes

Reviewer #2: No

4. Is the manuscript presented in an intelligible fashion and written in standard English?

Reviewer #1: Yes

Reviewer #2: Yes

5. Review Comments to the Author

Reviewer #1: General comments

This paper demonstrates that the approach towards defining regulatory areas matters for the fisheries resource within the regulatory area as fish stocks move through space.

I like the message here, it is concise and policy relevant. The boundary formation procedures here seem a little ad hoc. While defined by state or ports, the actual delineation is based on a GIS buffering approach. I suspect testing other approaches may be left for another paper, but for example this would be interesting to see using regulatory areas formed around historical use patterns by port as defined by the communities at sea data (Rogers et al., 2019 10.1038/s41558-019-0503-z) rather than a geometric buffering. The implications discussion is nice and linking it to the national standards is important - though the discussion there is centered around consequences of alternate approaches and design features of the permits. To balance this out, it would be nice to have a paragraph or so upfront talking about how regulatory boundaries are generally formed and potentially some alternate approaches. I’m thinking this could even draw from offshore oil and gas revenue approaches to revenue sharing (pool revenue across the entire area and kick back to states a proportion without drawing boundaries), the broad literature on gerrymandering, MARXAN style MPA optimization (and the analogous optimal dynamic solution framing of this problem) and so on. This would help clarify that this is a general concept and that there are a variety of groups thinking about these issues.

In-line comments

55-57: More broadly, managing to the extent of the environmental phenomenon reduces externalities and solution constraints for any environmental good/bad where there is market failure.

Figure 1: I think I’m having trouble understanding this figure. What are the colors referring to in step 3 (row 1)? How does step 3 translate into the result (row 1)? Is the “result” simply step 2 with the land masked out? Making this a bit clearer for the reader would be helpful.

Lines 278 - 279: All of the species seem to have some longitudinal shift in their centroid (Fig 2), perhaps this could be reworded to suggest that scup has the largest? Or perhaps longitudinal could be replaced with “further offshore” or “distance from land,” if that is what is meant?

Lines 307 - 308: Seems like black sea bass in CT or RI might be closer to zero change than summer flounder, based on Fig. 4

Lines 417 - 419: Permit holders may also engage in fishing practices that are short term profitable but harm the long term sustainability of the stock, to recuperate potential losses from shifting distributions. The private incentives for stewardship decrease.

Reviewer #2: This is a really terrific paper. My general suggestions are completely resolvable and are mostly suggestions to improve the paper. Naturally, my minor suggestions are all easily fixed. Except for the one major error I believe I have uncovered, described below, I would recommend this manuscript for publication following only minor revisions.

However, unfortunately, there is one major error, I believe, in the analyses that needs to be fixed. This will require redrawing several of the figures and potentially change the results and conclusions, unfortunately. It’s a trivial error, and easy to fix, but sadly it may have big consequences for the paper. Now maybe I’m wrong, I hope I am! But I need to be convinced that I am, because I have thought about this long and hard.

I believe the fishing port allocation analysis – the analysis to allocate landings to ports up to the 75% threshold of total landings - has been done incorrectly. Thinking about it logically, it doesn’t make sense that the number of total grid cells per state could decrease as the threshold of total landings increases, yet this is what is shown in Supp Fig S1, showing a sensitivity analysis of the 75% threshold versus other thresholds. It shows that as the threshold increases, the number of grid cells per state decreases, in every case. As a thought exercise, let’s say Ports A & B on average land 20% of the total landings, Port C lands 10%, and the rest of the ports land <10% each, adding up to 100% of landings. If you use a ranking system and apply a 50% threshold, then only Ports A-C would get an allocation. But if you apply a 75% threshold, then you start including ports with <10% landings historically, and the total number of ports receiving an allocation goes up. As you increase the threshold, you increase the number of ports contributing to that threshold. However, when you look at Supp FigS1, you can see that the number of grid cells allocated to each state actually *decreases* as the threshold increases. I think that all the analyses based on the fishing port allocation method need to be redone, including most of the figures.

When I go into the GitHub repo and read the R code for the port allocation method, it looks like the threshold ranking has been applied in the inverse. The notation in the code (lines 227-232) says that the top 75% of landings is extracted using “top_frac(0.25…) (line 231). But that code would actually extract the top 25% of landings, not the top 75% of landings, according to the R package documentation: https://dplyr.tidyverse.org/reference/top_n.html. This would imply that the results reported in the paper to reflect allocation of 75% of total landings instead reflect allocating only 25%. Likewise, the fractions used in the sensitivity analysis are all in the inverse, as well.

Once the analyses are redone, my concern with this threshold is the potential bias against states with lots of smaller ports. The sensitivity analysis is the right way to address this (Fig S1) but it will need to be redone with the right code.

Also, it doesn’t appear the data are contained in the GitHub repo – all the file pathways point to the lead author's computer filing system.

General suggestions

- Introduction

o I think a more powerful argument in general is that multiple management strategy shifts will be needed to address the challenge of range shifts. The argument in favor of changing methodologies for defining regulatory units and allocations does not depend on other approaches being faulty. “Another tool in the toolbox” is a stronger argument, I think. So I suggest rethinking the section that finds fault with dynamic management rules. The argument there seems to be about difficulty in defining management boundaries, a task also required in the approach suggested here, right?

o The paragraph starting on line 84 needs a bit of work. I think it’s meant to describe the intersection between spatial regulatory units and allocation in different situations. But the US example isn’t fleshed out – states have the rights to allocate fisheries within their borders, so the regulatory unit is state waters, but you don’t describe the allocation, i.e., to port, or community, as in Latin America. You don’t describe how federal water regulatory units are drawn or right allocated. Transitioning to the seasonal range shifts and migration portion of the paragraph is also a little awkward. Maybe the transition should be, “A challenge for all fixed spatial regulatory units used for allocating fishing rights is that fish distributions may change…” Also, given survey timing is one of the variables of interest in the present analysis, I suggest expanding on line 92-93, the point about seasonality being a key consideration of management, including how survey timing is related (or not) to seasonal distributions.

- Methods

o I suggest a paragraph at the beginning of the Methods section that describes the current management approach: how quota is set and allocated, using which data (years, seasons) are used, regulatory units, and allocation scheme. Moving the text from lines 170-173, for example, into such a paragraph would be helpful at the beginning of the section.

o Fish stock selection, lines 121-122: It is probably worth emphasizing the point that states allocate 100% of the fish stock in their state waters to their state’s vessels, and that it’s the allocation of fish stocks found in federal waters where regulatory units and state allocations really matter, and that is the focus on this paper. That would also eliminate the need to use “in federal waters” throughout. Though, certainly the point should be made that changing fish distributions will certainly also differently affect fisheries in state waters.

o Data used: Which years of landings data are used? I suggest stating which years of trawl survey data are used earlier in the paragraph than lines 166-167.

o Line 189: How were these years selected as the baseline? Did you do a sensitivity analysis to determine if the selection of those years influences the results?

- Results

- Please reconcile what appears to be a contradiction in results about seasonality. The paragraph starting on line 265 reports that seasonality is important in distribution, but the paragraph starting on line 305 reports that the consequences of seasonal differences for allocation are minor.

Minor suggestions

- Line 64: Suggest clarifying the meaning of this sentence by changing to either “have historically resulted” or “historically unsustainable.”

- Line 78: Regarding reason (i), in the present case, wouldn’t this reason be obsolete?

- Line 80: Change to “EEZs”

- Line 80-81: Clarify what is meant by “these boundaries.”

- Line 84: “Considering” is vague; can you change to “in association with” or something similar?

- Line 89: Suggest replacing “others” with “other considerations” or “other factors”.

- Line 108: By “different” do you mean “alternative”? Is one presently implemented and one is not?

- Lines 126-129: Make clear that it’s the landings from these historical periods upon which the quotas are based.

- Table 1: Summer flounder allocations don’t add up to 100%

- Lines 144, 148: Change “was” to “were”.

- Lines 217-219: Suggest using “biomass” instead of “trawl survey data”.

- Line 226: Suggest replacing “yearly-state estimation” with “year-state estimate”

- Lines 227-228: Suggest replacing “then pursue to compare” with “then compared”

- Line 226: Why go back as far as 1971 when the earliest data used to set quota for a stock were from 1980?

- Figure 2: Suggest using common names for the fish to match the text.

- Figure 3: It would be good to find a way to indicate state names on the figure, given you reference them in the text and there may be international or geographically-challenged American readers. Maybe a key showing just the outline of each state so readers can identify them by shape? Same comment as above about using the common name for the fish.

6. PLOS authors have the option to publish the peer review history of their article (what does this mean?). If published, this will include your full peer review and any attached files.

Reviewer #1: **Yes: **Robert Griffin

Reviewer #2: No

---

## [Author Response · Author response to Decision Letter 0]

13 Oct 2022

Dear editor and reviewers,

We would like to thank you for taking the time revising our manuscript. Please find attached a revised version of our manuscript and figures as well as a point-by-point responses to the comments raised by the reviewers. We have addressed all points raised by both reviewers and believe our paper has benefit from these changes.

Many thanks

---

## [Decision Letter · Decision Letter 1]

29 Nov 2022

Quantifying fish range shifts across poorly defined management boundaries

PONE-D-22-11430R1

Dear Dr. Palacios-Abrantes,

We’re pleased to inform you that your manuscript has been judged scientifically suitable for publication and will be formally accepted for publication once it meets all outstanding technical requirements.

Kind regards,

Caroline Ummenhofer

Academic Editor

PLOS ONE

Additional Editor Comments (optional):

Reviewers' comments:

Reviewer's Responses to Questions

**Comments to the Author**

1. If the authors have adequately addressed your comments raised in a previous round of review and you feel that this manuscript is now acceptable for publication, you may indicate that here to bypass the “Comments to the Author” section, enter your conflict of interest statement in the “Confidential to Editor” section, and submit your "Accept" recommendation.

Reviewer #1: All comments have been addressed

Reviewer #2: All comments have been addressed

2. Is the manuscript technically sound, and do the data support the conclusions?

Reviewer #1: (No Response)

Reviewer #2: Yes

3. Has the statistical analysis been performed appropriately and rigorously? 

Reviewer #1: (No Response)

Reviewer #2: Yes

4. Have the authors made all data underlying the findings in their manuscript fully available?

Reviewer #1: (No Response)

Reviewer #2: Yes

5. Is the manuscript presented in an intelligible fashion and written in standard English?

Reviewer #1: (No Response)

Reviewer #2: Yes

6. Review Comments to the Author

Reviewer #1: (No Response)

Reviewer #2: The authors have done an excellent job addressing my concerns and comments. I applaud their efforts, and am so happy that the solution was to revise wording and code commenting, and not results/conclusions!

7. PLOS authors have the option to publish the peer review history of their article (what does this mean?). If published, this will include your full peer review and any attached files.

Reviewer #1: **Yes: **Robert Griffin

Reviewer #2: **Yes: **Tessa B. Francis, PhD

---

## [Editor Report · Acceptance letter]

19 Dec 2022

PONE-D-22-11430R1 

Quantifying fish range shifts across poorly defined management boundaries 

Dear Dr. Palacios-Abrantes:

I'm pleased to inform you that your manuscript has been deemed suitable for publication in PLOS ONE. Congratulations! Your manuscript is now with our production department. 

Kind regards, 

on behalf of

Dr. Caroline Ummenhofer 

Academic Editor

PLOS ONE